# CoVT-CXR: Building Chain of Visual Thought for Interpretable Chest X-Ray Diagnosis

## Abstract

Though clinical report generation demonstrates the potential to improve the efficiency of radiologist workflow and benefits the under-served regions, automated analysis of radiographs suffers from un-interpretable progress and inaccurate results. To this end, we propose a novel Chain-of-Visual-Thought (CoVT) to emulate doctors' multi-modal reasoning, enabling more interpretable and accurate CXR diagnostic predictions with explicit multi-step intermediate guidance. Specifically, we mimic the multi-modal multi-step reasoning procedure of the doctors by breaking down clinical reports into individual descriptions and connecting each rationale to corresponding visual prompts—like masks, landmarks, linestrips, and bounding boxes—to illuminate the visual reasoning behind radiographs. By further dividing this association into cross-modal sub-tasks, CoVT is able to exploit a multi-stage fine-tuning protocol to gradually develop the chain-of-reasoning capability. To support this approach, we introduce CoVT-CXR, the first detailed-aligned, multi-step cross-modal dataset for diagnostic tasks, featuring about 3M instruction-following data points for pretraining and around 30K reasoning sequences for fine-tuning, sourced from 6K patient cases and annotated by 32 medical trainees using our tailored tool. Our CoVT-CXR covers more than 20 diseases, requiring 1 to 12 reasoning steps for diagnoses. Through a series of experiments on our CoVT-CXR, we demonstrate the advantages of the CoVT method over baseline approaches, validate the quality of our annotated data, and highlight the positive impacts of CoVT-CXR on various clinical-related tasks. Our CoVT model, annotation tool, and CoVT-CXR dataset will be fully available upon acceptance.

## 1 Introduction

Benefiting from the advanced comprehension capabilities of large language models (LLMs), Visual Language Models (VLMs) demonstrated significant achievements in common multimodal scenarios, such as automatic medical diagnostics, report generation, instruction-following, and image interpretation (Li et al., 2023e). Though showcasing impressive performances in general tasks, the vast majority of existing models, including GPT-4V, struggle with specialized medical imaging (Yang et al., 2023b; Wu et al., 2023). In particular, they often fail to find subtle pathologies, focusing instead on only inherent structures or prominent lesions.

Despite the initiative's attempts to replicate the success of GPT within the biomedical field by leveraging abundant datasets and large models (Li et al., 2023a; Singhal et al., 2023), these explorations neither yield the expected emergent intelligence nor provide explainable reasoning pathway in sophisticated medical scenarios, leaving a significant gap in diagnostic accuracy compared to human physicians. This observation is further explained by literature (Sucholutsky & Griffiths, 2023; Miller, 2019; Lake & Baroni, 2023) where they claim that the conventional end-to-end learning strategies demand massive data volumes and extensive training hours, leading to less efficient learning process compared to humans. Though incorporating intermediate human knowledge proves to be an effective solution (Guo & Bürger, 2022; Dhuliawala et al., 2023; Hong et al., 2023), research in medical scenarios is constrained not only by the inherently complex, multi-step, and cross-modal nature of medical reasoning but also by the scarcity of suitable datasets.

To this end, we introduce the very first multi-step cross-modal method for explainable clinical report generation, or **Chain of Visual Thought (CoVT)**. Unlike existing methods that rely on pre-defined

Figure 1: The chain of visual thought extends beyond plain reasoning to explore its visual dependencies, providing detailed grounding and guidance to form an intermediate diagnostic process.

reasoning steps (Pellegrini et al., 2023; Kougia et al., 2019) or single-step reasoning (Gu et al., 2024; Tanida et al., 2023), our CoVT enables more flexible and complex reasoning, resulting in improved performance and greater interpretability. To accomplish this, we break down the clinical report generation into multiple rationale description processes, where each description can be derived through several cross-modal reasoning steps. Taking the heart diagnosis in Fig.1 as an example, a strict process should follow: *identify heart structures, examine the heart-lung relationship, and measure the cardiothoracic ratio*, our CoVT is able to execute this protocol step-by-step with interleaved detailed visual examination, which is presented in the form of masks, landmarks, linestrips, and bounding boxes, and textual self-guided instructions to create an interpretable and traceable diagnosis. By further abstracting the cross-modal reasoning process into sub-tasks and implementing a multi-stage fine-tuning protocol, CoVT leverages the easy-to-hard spirit of curriculum learning (Hong et al., 2022; Azad et al., 2023) to progressively develop chain-of-reasoning capabilities.

To address the gap in detailed multi-step cross-modal dataset, we further introduce the first and foremost fine-grained and well-aligned cross-modal dataset based on MIMIC-CXR (Johnson et al., 2019), or **C**hain **o**f **V**isual **T**hought for **C**hest **X**-**R**ay (CoVT-CXR). Different from existing work that aims at simply increasing the sizes of datasets, our CoVT-CXR not only greatly enhances the interpretability of medical diagnostic tasks, which has been neglected in literature for a long time, but also enables novel designs in clinical-related tasks. In summary, our CoVT-CXR consists of $\sim$30k chain sequences from 6k CXR diagnostic cases, each of which is associated with various number of reasoning steps annotated by medical trainees. To simplify the annotation process and improve its efficiency, we introduce a tailored annotation tool, which not only allows fine-grained alignment across multiple modalities, but also enhances annotation efficiency by supporting semi-automated interactive curation in a human-in-the-loop manner. Please note that our CoVT-CXR is generic as all anatomical structures and major pulmonary pathologies are included.

To validate our hypothesis that CoVT yields more accurate and interpretable predictions, and that the CoVT-CXR dataset facilitates innovative yet interpretable designs for various medical tasks, we conduct comprehensive experiments on CoVT with the help of CoVT-CXR. By comparing prediction accuracy against several baselines, we demonstrate the overall effectiveness and the step-wise reasoning of CoVT. Additionally, we perform comprehensive ablation studies on CoVT, showcasing the necessity of the multi-step design and the effectiveness of our sub-tasks.

In all, our contributions can be summarized as follows:

- *CoVT-CXR dataset.* To the best of our knowledge, CoVT-CXR is the very first interpretable dataset for various CXR diagnostic tasks, thanks to its notable features where the well-aligned cross-modal reasoning process is annotated explicitly.

- *CoVT method.* We propose a novel CoVT method that allows more interpretable yet accurate CXR diagnostic predictions, demonstrating strong ability in various clinical-related tasks.

- *Open access.* Our dataset, code as well as the tool will be fully public upon acceptance.

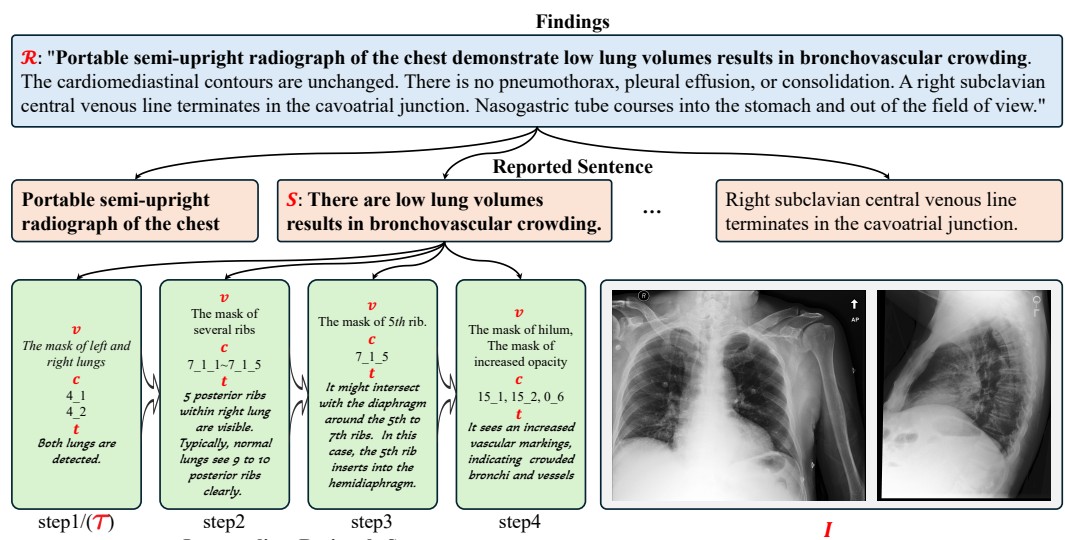

Figure 2: An example annotation for one $\mathcal{R}$. We highlight only the annotations for one $S$ in red.

## 2 DATASET CURATION

**Motivation.** As described in previous sections, the ultimate goal of our dataset CoVT-CXR is to shed light on intermediate reasoning steps from well-trained doctors, thereby encouraging research into designing interpretable yet accurate methods for diagnostic tasks. In this paper, we formulate this reasoning procedure into a multi-step, cross-modal pipeline that not only mimics the underlying reasoning path of doctors but also explicitly bridges the gap between the input CXR image and the generated report. Assuming doctors are provided with a CXR image, they will first take a glance at the radiography and roughly separate various lobes along with their semantics. Initial reasoning based on the segmented results is conducted. This initial reasoning is further utilized to identify Regions of Interests (RoIs) and guide more detailed measurements on top of them, together with the chief complaint of this patient when available. The reasoning for individual RoIs that belong to the same lesion or structure is grouped together, resulting in a single interpreted sentence for each. When multiple lesions or structures are present, this process culminates in a comprehensive report for this CXR. Motivated by this, we request the annotators to follow the widely adopted 'ABCDE' approach[1] (A-Airway; B-Breathing; C-Cardiac; D-Diaphragm; E-Everything else) to ensure a comprehensive interpretation. And clinical logic flow is roughly based on the formal diagnostic guidelines outlined at radiology masterclass[2]. All medical trainees present their reasoning steps sequentially in the form of textual, visual, and cross-modal annotations. In the following paragraphs of this section, we will elaborate on the metadata collection and our annotation tool.

**Metadata Collection.** During the annotation process, the annotators are provided with a CXR image $I$ as well as its original comprehensive report $\mathcal{R}$ from an existing dataset. They are asked to first decompose $\mathcal{R}$ into report segments $S \in \mathcal{R}$, then figure out the corresponding set of visual points $v \in \mathcal{P}$ and its semantics $c \in \mathcal{C}$ described by this $S$. Finally, an intermediate text description $t$ is also requested to explicitly represent the underlying prior knowledge for each $v$ and $c$. Specifically, $\mathcal{P}$ denotes the $x, y$ co-ordinate space and $\mathcal{C} = \{1, \ldots, 112\}$, reflecting pre-defined 112 semantic classes.

Mathematically, we have metadata defined as $\mathcal{T} = \langle I, S, v, t, c \rangle$. Please note that in $\mathcal{R}$, natural decomposition according to punctuation can often be confusing. For example, two sentences might describe the same structure, or a single sentence might refer to multiple lesions. To address these entangled interpretations, we propose decomposing or merging every diagnostic sentence. Specifically, any report sentence that corresponds to more than one lesion or structure will be split, while sentences referring to the same lesion or structure should be merged. This instruction guides the creation of our $S$. Subsequently, annotators will present their own diagnostic reasoning to explicitly detail the

---

[1] https://geekymedics.com/chest-x-ray-interpretation-a-methodical-approach/
[2] https://www.radiologymasterclass.co.uk

| Dataset | Subjects | $C_n$ | Det. | Seg. | VQA | Gen. | Rat.[3] |
|---|---|---|---|---|---|---|---|
| JSRT (Shiraishi et al., 2000) | BBox | 2 | ✓ | ✗ | ✗ | ✗ | ✗ |
| MC (Jaeger et al., 2014) | Polygon | 2 | ✓ | ✗ | ✗ | ✗ | ✗ |
| SH (Jaeger et al., 2014) | BBox | 2 | ✓ | ✗ | ✗ | ✗ | ✗ |
| ChestX-ray8 (Wang et al., 2017) | BBox, Description | 8 | ✓ | ✓ | ✗ | ✗ | ✗ |
| ChestX-ray14 (Wang et al., 2017) | BBox, Description | 14 | ✓ | ✓ | ✗ | ✗ | ✗ |
| IU X-Ray (Kougia et al., 2019) | Description | - | ✓ | ✗ | ✗ | ✓ | ✗ |
| CheXpert (Irvin et al., 2019) | Description | 14 | ✓ | ✗ | ✗ | ✓ | ✗ |
| MIMIC-CXR (Johnson et al., 2019) | Description | - | ✓ | ✓ | ✗ | ✓ | ✗ |
| PadChest (Bustos et al., 2019) | BBox, Polygon, Description | 174 | ✓ | ✓ | ✗ | ✗ | ✗ |
| VinDr-CXR (Nguyen et al., 2022) | BBox, Polygon, Description | 28 | ✓ | ✓ | ✗ | ✗ | ✗ |
| MS-CXR (Boecking et al., 2022) | BBox, Description | 8 | ✓ | ✗ | ✗ | ✗ | ✗ |
| CTR-CPAR (Duvieusart et al., 2022) | BBox, Polygon, Table | 2 | ✓ | ✓ | ✗ | ✗ | ✗ |
| Medical-CXR-VQA (Hu et al., 2024b) | Description | 35 | ✓ | ✗ | ✓ | ✗ | ✗ |
| LLM-CXR (Lee et al., 2024) | BBox, Description | - | ✓ | ✓ | ✓ | ✓ | ✗ |
| **CoVT-CXR** **(Ours)** | BBox, Polygon, Landmark Linestrip, Description | 112 | ✓ | ✓ | ✓ | ✓ | ✓ |

Table 1: Comparison of our CoVT-CXR with mainstream datasets. $C_n$, Det., Seg., VQA, Gen., and Rat. represent the number of classes in each dataset, and whether they involve detection, segmentation, visual question answering, generation, and rationale, respectively..

Figure 3: The visual cue class[4] distribution of all identified lesions and anatomical structures during diagnosing, which can be represented through masks, linestrips, or landmarks.

intermediate steps for each $S$, ensuring that the content is not interwoven. This process will produce well-aligned $v$, $c$, and $t$ for each specified intermediate step. Thanks to our design, the entire dataset features fine-grained multi-modal alignment and a multi-step structure. We provide an example in Fig. 2 and highlight our annotations in red. As shown in this figure, given the original report $\mathcal{R}$, the annotator first decomposes it into multiple sentences $S$. For each individual $S$, several metadata are further annotated, with the number of $\mathcal{T}$ corresponding to the required intermediate steps for that $S$. Specifically, we have 4 and 6 $\mathcal{T}$ under the highlighted $\mathcal{S}$ and for the entire $\mathcal{R}$, respectively. The forms of $v$, $c$, and $t$ may vary at each intermediate step. For example, $c$ could be 4_1/4_2 and 7_1_5 in steps 1 and 3, meaning *both lungs* and *5th right rib* semantic classes. while $t$ is *Both lungs are detected* in step 1. Please note that $v$ are in the form of masks for all $\mathcal{T}$, but they can take different forms in practice. We refer the readers to the Appendix for more annotation examples.

As a result, we ask 32 medical trainees and construct a very comprehensive CXR dataset including 6k cases, 10k radiographs, and 70k metadata descriptions from 30k reported interpretations. Fig.3 provides an overview of our collected dataset. We refer the readers to our Appendix for more dataset statistics.

**Annotation tool.** As shown in $\mathcal{T}$, our annotations offer a comprehensive view of the entire dataset. However, this process is extremely time-consuming for annotators. Consequently, no mainstream applications (Wada; CVAT.ai Corporation, 2023; AIPair, 2023) are capable of building such complex multi-modal datasets. To this end, we have modified an enhanced version of the open-source Labelme (Wada) into a customized annotation platform, Labelme-CoVT, tailored to match the schema set forth by CoVT-CXR. This tool allows annotators to select different perspective images, decompose report sentences $\mathcal{R}$ as needed, and provide metadata $\mathcal{T}$, including the corresponding category $c$, mask $v$, and description $t$, for each step. Additionally, with the integration of AI models, Labelme-CoVT boosts annotation efficiency by supporting semi-automated interactive curation in a human-in-the-loop manner. More features of Labelme-CoVT can be found in the Appendix. Our CoVT-CXR, Labelme-CoVT, and AI model integrations will be publicly available upon acceptance to foster the broader creation of multi-modal chain reasoning data within the community.

**Comparison with existing chest X-ray dataset.** As shown in Tab. 1, Our CoVT-CXR is the first dataset specifically designed to offer multi-step cross-modal annotations, an aspect that has been neglected yet is crucial for the field. In contrast to existing datasets, ours features greater diversity, incorporating a wider array of annotation types and supporting multi-class annotations. This variety greatly enhances the interpretability and accuracy of designs for various clinical tasks.

---

[4]CoVT-CXR contains many ribs labeled to help describe relevant lesions, even though MIMIC-CXR has reported fewer rib lesion cases. Detailed discussions about ribs are provided in the Appendix.

[3]**Rat.** refers to multimodal rationale, requiring reasoning with both visual and textual content. Therefore, datasets that include only textual chains of thought are not included.

**Multi-task Overview**

$\mathcal{R}$: PA and lateral radiographs of the chest were acquired. The lungs are clear, but hyperinflated. The heart size is normal. The mediastinal contours are normal. There are no pleural effusions. No pneumothorax is seen.

**T1: Segment Anything**
*<Input>:*
<classes>/<point>
*<Output>:*

**T2: Interactive image caption**
*<Input>:*
*What is the findings in highlighted region?*
*<Output>:*
*The highlighted region indicates a normal mediastinum with no visible abnormalities.*

**T3: Instruction-following visual reasoning**
*<Input>:*
*<Measure the width of the heart>*
*<Output>:*

**T4: Visual question answering**
*<Input>:* *<Output>:*
*What is the condition of the lungs?*
*The lungs from frontal view appear within normal. There are no pneumothorax.*
*lateral view also show clear cavity, and unblunted costophrenic angles suggest no pleural effusion.*
*<findings> The lungs are within normal, there are no pleural effusions and pneumothorax is seen. </findings>*

**T5: Explainable report generation**
*<Output>:*
*PA and lateral views of the chest.*
*The lungs from frontal view appear within normal. There are no pneumothorax.*
*lateral view also show clear cavity, and unblunted costophrenic angles suggest no pleural effusion.*
*Total 10 right ribs and 11 left ribs visible, and all of them seems intact.*

*8 anterior ribs present in the lungs, with 7th inserts into the hemidiaphragm. more than 7 anterior ribs indicates lung hyperexpansion.*
*Heart detected and appear to stable.*
*The CTR is 40.9%, the heart size is normal*
*The mediastinal structure and contours are normal.*

Figure 4: Illustration of five tasks derived in an easy-to-hard manner for CoVT.

## 3 OUR METHOD

Given a CXR image $I$, our goal is to generate multiple $S$s such that the final clinical report $\mathcal{R}$ is the set of all generated $S$. This task appears manageable with the assistance of our annotation $\mathcal{T}$ for each decomposed sentence $S$ in CoVT-CXR, as it's feasible to achieve this goal by integrating the intermediate reasoning steps from doctors into diagnostic tasks using the provided $\mathcal{T}$. However, we would like to argue that managing various steps of cross-modal reasoning is too complex to be effectively trained in an end-to-end manner (See Sec. 4 for results), therefore, specialized designs are required. In this section, we will describe our designs of CoVT by outlining how to decompose the reasoning steps in an easy-to-hard manner and structure $\mathcal{T}$ accordingly, describing how the modality-unified representation in CoVT is designed, and detailing our multi-stage training protocol.

### 3.1 STEP DECOMPOSITION

As outlined in our motivation, doctors are skilled in performing multi-step reasoning based on their observations and measurements of CXR images, with or without a chief complaint. The step decomposition aims to break down the reasoning steps and organize $\mathcal{T}$ accordingly (see Fig. 4 for an overview). As depicted in this figure, our decomposition not only mirrors the underlying reasoning process of doctors, but also allows for extensions to various clinical-related tasks.

**T1: Segment anything for CXR.** Upon receiving a CXR, doctors swiftly recognize various points, lines, and regions along with their meanings. Inspired by this, the initial step is framed as an **image-to-image** task constructed from metadata, which can be represented as $\langle I, p, v' \rangle \rightarrow v$. Here, $v'$ is initially empty ($\emptyset$), but can inherit from $v$ subsequently. $p$ denotes the prompt under the Segment Anything (SAM) (Kirillov et al., 2023) framework for CXR image, where it either corresponds to $c$ or can be points sampled from $v'$. In contrast to existing approaches that either perform unsupervised segmentation without semantic knowledge or only segment predefined classes, we argue that a comprehensive model should be capable of capturing all lesions and structures in CXR images. This

provides valuable prior knowledge for downstream tasks and lays the foundation for more advanced reasoning. To achieve this, we adopt models (Kirillov et al., 2023; Zou et al., 2023; Xiong et al., 2023) trained on large-scale natural images, while employing prompt engineering to enable zero-shot capability for medical images. However, even models fine-tuned with medical images (Ma et al., 2024) struggle to capture overlapping structures like bones. This further motivates us to explore SAM framework. We train the SAM-CXR model from scratch, aiming to identify potential targets in chest X-rays. For data engineering, we refer to (Kirillov et al., 2023), initially organizing a large-scale manually labeled dataset to develop the early-stage model. We then expand the dataset using a semi-automated human-in-the-loop annotation approach. The SAM-CXR model is subsequently used to accelerate downstream data curation and pretrain.

**T2: Interactive image caption.** Upon receiving initial visual cues from **T1**, doctors proceed to draw conclusions or summarize these results. Typically, these masks highlight regions in the CXR image, including morphology, size, and quantity. Meanwhile, the conclusion or summary is represented in text form, which contrasts with the highly abstract nature of masks. This text further describes the features of textures, measurements, and anomalies within the masked regions, which can be difficult to convey visually. To address this, we formulate this second step as an interactive **image-to-text** generation task. Mathematically, we have $\langle I, p, v \rangle \rightarrow t$, where $p$ has the same definition as in **T1**. The objective of **T2** is to receive masks in the form of points, classes, or masks, and then generate interpretable and well-aligned reports for these specified regions. In doing so, we draw upon existing work on image captioning with region guidance (Huang et al., 2023; Lai et al., 2023; Huang et al., 2024). Compared to similar works lacking visual prompts, these approaches offer greater flexibility in terms of interactivity and controllability.

**T3: Instruction-following visual thought.** Upon receiving the internal description $t$, the doctor may reference the chief complaint or follow typical instructions to identify Regions of Interests (RoIs) and then conduct more detailed measurements, resulting in multiple visual cues. Therefore, we frame this step as a **text-to-image** generation task, or $\langle I, q \rangle \rightarrow \langle v_1, \ldots, v_i \rangle$ in mathematical terms. Here, $q$ represents the instruction that doctors wish to follow, sourced from either the chief complaint or predefined template instructions available in the dataset. For instance, if we denote our $q$ as *"Assessing ET tube's position"* (Johnson et al., 2019), the objective of **T3** is to spontaneously obtain visual cues for the ET tube, the tip of the tube, the carina, and potentially the trachea, along with distance measurements. To accomplish this, we draw inspiration from prompt-guided CoT reasoning in NLP tasks (Wei et al., 2022; Yao et al., 2023b; Besta et al., 2024b), where logical reasoning abilities are significantly enhanced, leading to notable improvements in intelligence. While VLMs equipped with cross-modal CoT demonstrate their multi-modal capabilities, nearly all purported multi-modal CoTs (Chen et al., 2024; He et al., 2024) fail to achieve inherent multi-modality. This is primarily because these models passively accept multimodal inputs, allowing CoT reasoning to remain confined to a single modality within the language landscape. In contrast, our **T3** focuses on constructing visual CoT sequences to imbue models with intrinsic cross-modal CoT reasoning capabilities.

**T4: Visual question answering.** At this step, doctors group related visual cues to generate one round of reasoning. For example, visual cues belonging to one structure or lesion are parsed and reasoned together to conclude one $S$. Formally, the task aims to obtain data in the form $\langle I, q' \rangle \rightarrow \langle (v_1, t_1), \ldots, (v_i, t_i), S \rangle$, where each visual cue $v_i$ is coupled with its corresponding text description $t_i$. Our **T4** concludes with a generated description $S$ as the answer to $q'$. A typical $q'$ might be *"What is the condition/finding of a specific field"*. In practice, $q'$ is obtained from GPT, where questions are generated based on the textual content $S$ during training and then applied to all test CXR images. Our design aligns with the data workflows of LLava (Liu et al., 2023a) and LLaVa-Med (Li et al., 2023a) (see Appendix) where questions serve as instructions(Lee et al., 2023). Not surprisingly, it can be also regarded as medical Visual Question Answering (VQA) task (Pellegrini et al., 2023; Moon et al., 2022). Notably, this task focuses on **single-round** questioning based on individual sentences in the report, disregarding other issues present in the CXR image.

**T5: Explainable report generation.** Single-round reasoning serves as the fundamental component for our final step, which is our ultimate goal of explainable report generation, or CoVT. Specifically,

$$I \rightarrow \left\langle \begin{matrix} (v_1^1, t_1^1), & \ldots, & (v_{i_1}^1, t_{i_1}^1), & S^1, \\ & & \vdots & \\ (v_1^M, t_1^M), & \ldots, & (v_{i_M}^M, t_{i_M}^M), & S^M, \end{matrix} \right\rangle,$$

where the final report $\mathcal{R}$ consists of $M$ sentences in total. $(v_{i_j}^m, t_{i_j}^m)$ is the paired data composing of visual cue $v_{i_j}^m$ and text cue $t_{i_j}^m$ at the $i_j$-th step for sentence $S^m$, where $m \in \{1, \ldots, M\}$. Equivalently, we can reformulate **T5** as $I \to \{\langle \{v_{i_j}^m, t_{i_j}^m\}_{i_j=1}^{i_m}, S^m \rangle\}_{m=1}^M$.

Clearly, **T5** represents an open-ended task involving **multiple-round** sentence generation. It demands the model to autonomously identify and report all relevant targets within the images, regardless of the presence of lesions or prompts. To accomplish this, the step endows the model with the capability to conduct step-by-step automated diagnosis while offering the rationale for each intermediate step, thereby achieving explainable report generation.

## 3.2 MODALITY-UNIFIED DATA REPRESENTATION

In this section, we will introduce our CoVT. To prevent the introduction of redundant encoders caused by dynamic data structures, we employ a unified representation learning approach that yields modality-agnostic input. Unlike (Boecking et al., 2022; He et al., 2022; Oquab et al., 2023), which integrate reconstruction and semantic understanding to achieve excellence across diverse tasks, our approach prioritizes lossless reconstruction alone. Following the method in (Razavi et al., 2019), we use variational autoencoders to discretize image features, enabling the integration of visual cues and discrete textual symbols. Given the fine-grained nature of medical image representations, especially concerning textures related to abnormal diagnostic signs, we utilize VQ-GAN (Esser et al., 2021) to compress detailed features.

Given a predefined dictionary quantizer $\mathcal{Q}(\cdot)$, we begin with sampling individual visual cue $x$ from either the medical image set $\mathcal{I} = \{\bigcup I\}$ or the visual sets $\mathcal{V} = \{\bigcup v\}$. By discrete quantization, we obtain a representation codebook by compressing and subsequently reconstructing the visual cues, which is obtained by minimizing the following equation:

$$\mathcal{L}_{\text{GAN}}(\mathcal{E}, \mathcal{D}, \mathcal{Q}) = \|x - \hat{x}\|^2 + \|\text{sg}[\mathcal{E}(x)] - \mathcal{Q}(\mathcal{E}(x))\|_2^2 + \beta \|\text{sg}[\mathcal{Q}(\mathcal{E}(x))] - \mathcal{E}(x)\|_2^2. \quad (1)$$

where $\text{sg}[\cdot]$ denotes the gradient-stop function, and the constructed image $\hat{x}$ is obtained by $\mathcal{D}(\mathcal{Q}(\mathcal{E}(x)))$. In particular, $\mathcal{E}(\cdot)$ and $\mathcal{D}(\cdot)$ are encoder and decoder of VAE (Esser et al., 2021) respectively. $\beta$ is the hyper-parameter. Subsequently, we merge the visual representation codebook with the text dictionary of LLMs to construct a modality-unified tokenizer. This design allows input from visual and/or textual cues to be encoded into discrete tokens, facilitating further processing by LLMs.

## 3.3 TRAINING PROCEDURE OF COVT

Inspired by curriculum learning (Gao et al., 2024; Azad et al., 2023; Li et al., 2023c), which shares a similar philosophy to our easy-to-hard task decomposition, we propose a multi-stage fine-tuning protocol to gradually cultivate the chain-of-reasoning capability of our CoVT model. The data volume for multi-stage training can be found in the Appendix.

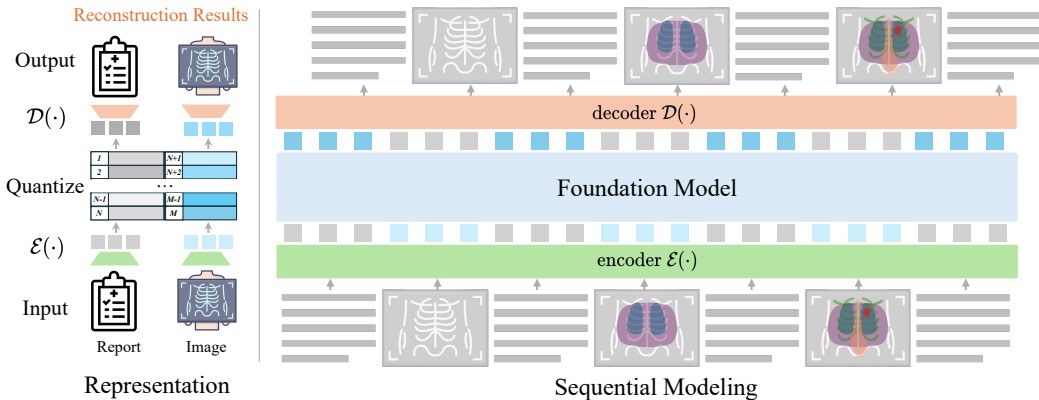

Figure 5: The architecture with sequential modeling and quantized representation for building CoVT.

**Step-agnostic sequential modeling.** While decomposing the process into multiple steps, training specialized models for each step is inefficient. Instead, we aim for a unified model capable of achieving multiple objectives from these steps. Therefore, we adopt the concept of In-Context Learning (ICL) to establish a task-agnostic model, or in our context, a step-agnostic model.

Inspired by (Bai et al., 2023b), we adopt the sequential modeling for multi-task training. Taking the **T4**, or the VQA step, as an example, we concatenate the input-output pair of the sampled data, restated as $\langle I, q, v_1, t_1, \ldots, v_i, t_i, S \rangle$. As described in Sec. 3.2, the textual and the visual cues are tokenized through off-the-shelf textual tokenizers and the pre-trained autoencoder according to Eq. 1 respectively, as shown in Fig.5. Then they are merged and flattened into a one-dimensional sequence, namely, $d_x = [d_I, d_q, d_{v_1}, d_{t_1}, \ldots, d_{v_i}, d_{t_i}, d_S]$. Specifically, $d_x \in \mathbb{R}^D$ has a length of $D$ and $d_{x,s}$ denotes the value of $d_x$ at location $s = \{1, \ldots, D\}$. Then our model is trained in an autoregressive manner by minimizing the following objective function:

$$\mathcal{L}_{\text{AR}}(\theta) = -\sum_{s=1}^{D} \log P(d_{x,s} \mid d_{x,s'<s}; \theta) \tag{2}$$

We would like to highlight that sequential modeling is very well-suited to our CoVT since it enables complete sequence training instead of an ensemble Mixture of Experts (MoE) (Shazeer et al., 2017), where the error accumulation resulting from out-of-distribution intermediate results is a common occurrence and can be further propagated and amplified along the chain of reasoning.

**Multi-stage fine-tuning of CoVT.** Rather than directly targeting **T5** from the outset, we advocate for adopting a multi-stage fine-tuning strategy, inspired by curriculum learning. This strategy follows an easy-to-hard paradigm, which often yields better outcomes. Intuitively, simpler tasks can serve as stepping stones toward mastering more sophisticated ones by leveraging variations in task complexity. To strive for a balance between curriculum learning (Hong et al., 2022) and catastrophic forgetting (Ramasesh et al., 2022), our learning process consists of both training and fine-tuning stages. In the initial training stage, the focus lies on equipping the model with the ability to comprehend raw medical images and the described textures associated with these images. Naturally, we categorize the first three steps (**T1**-**T3**) within this stage. It's worth noting that although **T3** involves intricate reasoning due to its multi-step visual thought process, we include it in the training stage as it involves only one modality in its prediction during inference. During the subsequent fine-tuning stage, the model utilizes interleaved data from tasks **T4** and **T5** to develop its multi-step chain reasoning capabilities. The primary objective is to enable the trained model to generate text integrated with images while maintaining the coherence of the generated content.

## 4 EXPERIMENT

We conduct comprehensive experiments on CoVT-CXR dataset, demonstrating the superiority of our proposed CoVT method over existing baselines. Our results further highlight the necessity of introducing multi-step cross-modal reasoning and its positive impact on various clinical-related tasks.

**Experimental details** We divided our CoVT-CXR dataset into two non-overlapping subsets: 5.6k patient cases for training and 400 patient cases for testing. Unless otherwise specified, the training set is used for model training and fine-tuning, and all results are reported based on the test set. We compare to various types of methods, including zero-shot methods (Liu et al., 2023a; Lee et al., 2024; Abdin et al., 2024), few-shot models (Yang et al., 2023b; Team et al., 2024), and models that are fine-tuned with our CoVT-CXR (Abdin et al., 2024; Liu et al., 2023a). We follow the existing work (Lee et al., 2023) to setup our evaluation metrics. We refer the readers to the Appendix for more details about our hyper-parameters, computational resources, and other implementation details.

### 4.1 CLINICAL REPORT GENERATION RESULTS ON CoVT-CXR

Due to space constraints, we only report the CoVT results for **T4** and **T5** in Tab. 2 in the main paper. Results for **T1**-**T3**, more evaluation metrics, and visual examples are available in the Appendix. For **T4**, our CoVT consistently achieves the best performance across all settings, even without visual pre-training. This observation reinforces our claim that CoVT is a unified and effective model for multiple clinical-related tasks. We would like to emphasize that our well-aligned cross-modal

| Strategy | Model | Params | BLEU-1 | BLEU-2 | BLEU-4 | ROUGE-1 | ROUGE-L | METEOR | CIDEr |
|---|---|---|---|---|---|---|---|---|---|
| **Visual Question Answering** | | | | | | | | | |
| Few-shot | GPT-4V | - | 0.148 | - | 0.040 | - | 0.246 | - | 0.238 |
| | GPT-4o | - | 0.116 | 0.066 | 0.028 | 0.235 | 0.198 | 0.269 | 0.080 |
| | Gemini-1.5Pro | - | 0.226 | 0.156 | 0.081 | 0.369 | 0.339 | 0.361 | 0.596 |
| Finetuned | LLaVA-1.5 | 7B | 0.508 | 0.449 | 0.358 | 0.659 | **0.649** | 0.563 | 3.103 |
| | Phi-3V | 3.8B | 0.508 | 0.450 | 0.361 | 0.659 | 0.648 | **0.565** | 3.233 |
| | **CoVT** | 7B | **0.530** | **0.466** | **0.370** | **0.660** | 0.649 | 0.561 | **3.539** |
| **Report Generation** | | | | | | | | | |
| Zero-shot | LLaVA-1.5 | 7B | 0.096 | 0.042 | 0.009 | 0.163 | 0.123 | 0.144 | - |
| | LLM-CXR | 3B | 0.011 | 0.005 | 0.002 | 0.063 | 0.056 | 0.025 | 0.006 |
| | Phi-3V | 3.8B | 0.091 | 0.028 | 0.007 | 0.147 | 0.102 | 0.106 | 0.001 |
| Few-shot | GPT-4V | - | 0.099 | 0.046 | 0.012 | 0.212 | 0.133 | 0.224 | - |
| | GPT-4o | - | 0.092 | 0.043 | 0.012 | 0.217 | 0.134 | 0.224 | - |
| | Gemini-1.5Pro | - | 0.207 | 0.105 | 0.042 | 0.332 | 0.220 | 0.229 | 0.044 |
| Finetuned | LLaVA-1.5 | 7B | 0.319 | 0.233 | 0.133 | **0.517** | **0.459** | 0.351 | 0.054 |
| | Phi-3V | 3.8B | 0.317 | 0.229 | 0.128 | 0.503 | 0.449 | 0.341 | 0.165 |
| | **CoVT** | 7B | **0.341** | **0.245** | **0.135** | 0.500 | 0.434 | **0.354** | **0.330** |

Table 2: Performance comparison of report generation based on CoVT-CXR

CoVT-CXR dataset can provide significant additional benefits, besides demonstrated five tasks, to the community.

We present our report generation results in Tab. 2. Once again, our CoVT consistently delivers the best performance among all fine-tuned models, outperforming state-of-the-art methods across 5 out of 7 evaluation metrics. Notably, our model achieves significant improvements in CIDEr scores, with relative gains of 100% and 511% over Phi-3V and LLaVA 1.5, respectively. As expected, zero-shot methods lag behind few-shot methods, and few-shot methods are not on par with fine-tuned models. Among the closed-source models, Gemini shows markedly higher generative capability for CXR diagnostic tasks, often performing twice as well as GPT. Overall, un-finetuned models struggle with effectively reasoning on our CoVT-CXR dataset, highlighting the complexity of both the dataset and the task. In contrast, by exploiting cross-modal chain-of-thought reasoning, our CoVT is able to generate superior results over the complex report generation task, even without visual pre-trained.

## 4.2 Ablation Studies on CoVT

We report several key ablations refer the readers to the Appendix for more.

| Model | $N_{step}$ | BLEU-4 | ROUGE-L | METEOR | CIDEr |
|---|---|---|---|---|---|
| Bliva | 0 | 0.064 | 0.131 | 0.142 | 0.076 |
| | 1 | **0.153** | **0.272** | **0.191** | **0.281** |
| Phi-3V | 0 | 0.128 | 0.449 | 0.341 | **0.165** |
| | 1 | **0.138** | **0.462** | **0.355** | 0.140 |
| **CoVT** | 0 | 0.125 | 0.417 | 0.329 | 0.212 |
| | 1 | 0.130 | 0.426 | 0.338 | 0.239 |
| | $n$ | **0.145** | **0.434** | **0.354** | **0.330** |

Table 3: The impact of the number of steps on CoVT.

**Is multi-step reasoning really necessary?** We introduce a plain CoVT as our zero-step baseline such that $I \to \langle \{S^m\}_{m=1}^M \rangle$. Grounding-based VLMs (Abdin et al., 2024; Hu et al., 2024a) is selected as our one-step reasoning baseline. And both methods are finetuned on CoVT-CXR. Results in Tab. 3 support our claim that both one-step VLM and CoVT benefit from step-wise reasoning, showcasing that multi-step reasoning is necessary.

**Can we benefit from long chains?** To determine whether extended reasoning leads to better performance, we conducted an experiment where the top $[0, 30, 60, 100]$ percent of the ground truth intermediate results are fed to CoVT, followed by subsequent reasoning. The results, shown in Fig. 6, indicate that CoVT's accuracy is proportional to the extent of the reasoning process it accesses, suggesting that more intermediate steps help and our annotations are of high quality.

## 5 RELATED WORK

**Vision Language Models.** Recent advancements in AI have led to significant progress in Vision Language Models (VLMs), which demonstrated remarkable performance in various tasks, including conversational AI (Wang et al., 2023b; Bai et al., 2023a; Lu et al., 2024), image captioning (Li et al., 2023d; Koh et al., 2023; Yang et al., 2023a) and comprehensive world knowledge understanding (Peng et al., 2023). Early VLMs, such as miniGPT4 (Zhu et al., 2023), LLaVA(Liu et al., 2023b), and LLaVA 1.5 (Liu et al., 2024), were primarily designed to process single images and were subsequently fine-tuned for specific domain applications, such as medical image understanding (Omkar Thawkar, Abdelrahman Shaker, Sahal Shaji Mullappilly,

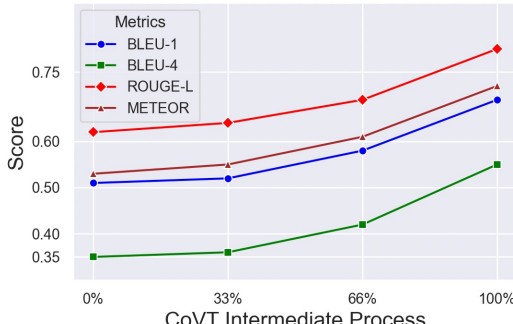

Figure 6: As the amount of information from intermediate steps increases, CoVT demonstrates significant performance boosts.

2023; Lee et al., 2024; Li et al., 2023b; Moor et al., 2023). Moreover, native multimodal models, such as GPT-4V (OpenAI et al., 2024), Med-Gemini (Saab et al., 2024), Emu, and Emu2 (Sun et al., 2023b;a), can accept multimodal inputs and generate multimodal outputs, showcasing their capability to integrate and process various modalities of information simultaneously. While these native multimodal models primarily establish mappings among images, audio, and text (Yin et al., 2023), our method builds the sequential interleaved rationale with both semantic visual cues and textual prompts, offering greater extrinsic interpretability.

**Multimodal Chain-of-Thought.** Chain-of-Thought (CoT) (Wei et al., 2023) reasoning boosts model interpretability by breaking down intricate tasks into manageable steps, thus improving performance in multi-step tasks such as logical inference (Chu et al., 2023). Tree-of-Thought (ToT) (Yao et al., 2023a) employs a hierarchical structure, providing it more optional sampling trajectories for self-verification. Graph-of-Thought (GoT) (Besta et al., 2024a) further expands ToT toward graph reasoning to build crisscross dependencies. Although CoT was initially derived in text-only language models, it also can be immensely useful in visual tasks, such as visual question answering, multimodal editing, or visual captioning (Lu et al., 2022; Zheng et al., 2023; Wang et al., 2023a; Gupta & Kembhavi, 2022; Himakunthala et al., 2023). However, current so-called multimodal CoT techniques consider multimodal inputs but rely solely on text for generation (Hu et al., 2024b), overlooking that text alone fails to fully capture cognitive processes, such as performing medical measurements (Duvieusart et al., 2022). Our approach introduces versatile visual prompts in the multimodal chain-of-thought, significantly enhancing the explainability and traceability of the models.

## 6 LIMITATION

Notably, the extended context significantly increases both training and inference costs. However, as a sparse modality, visual cues may not require full-size image tokens, suggesting that adaptive representation learning (Duggal et al., 2024) could help reduce context length. A shorter chain facilitates historical case comparison like Maira-1/2 (Hyland et al., 2023; Bannur et al., 2024). Furthermore, the interleaved reasoning model we implemented is relatively simple. Current LLMs/VLMs lack the capability for image generation, highlighting the need for specialized models. Finally, while we demonstrated the potential of CoVT in CXR diagnosis, errors in intermediate steps could undermine overall performance. We hope that CoVT-CXR contributes to advancing related research and inspires new studies to address these challenges.

## 7 CONCLUSION

In this paper, we introduce a well-aligned cross-modal medical dataset CoVT-CXR where the intermediate reasoning steps are annotated explicitly. Based on this, we further propose a novel CoVT method that can incorporate these intermediate steps into predictions, leading to more interpretable yet accurate results. Our dataset, method, and annotation tool will be fully available upon acceptance.

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
