# SUPPLEMENTARY MATERIAL

As described in our main paper, we will provide more details about our dataset CoVT-CRX, annotation tool Labelme-CoVT, prompt generation process, and implementation details in Sec. A, Sec. B, Sec. C, and Sec. D, respectively. Results on task **T1**, **T2**, and **T3** are provided in Sec. E, proving the multi-task ability of our CoVT method on various clinical-related tasks. Finally, we refer the readers to Sec. F for visual examples of our CoVT results on **T5**. Our code, data, demo, and annotation tool are partially available in this anonymous link and will be fully accessible upon acceptance.

## A    DATASET DETAILS

Besides the visualization in our main paper, we provide more quantitative measures on CoVT-CXR, including word frequency of description, the pre-defined semantic label space $\mathcal{C}$ as well as the statistics at each step. Fig. 1 offers a striking word cloud visualization. In part (a), we see findings like cardiac silhouette and hilum contours, depicted through points, lines, and masks. In part (b), common pathologies stand out, with terms such as 'enlargement,' 'low,' 'mild,' and 'crowding' frequently appearing. These subtle gradations are particularly elusive for current vision-language models, making them all the more fascinating and challenging to interpret.

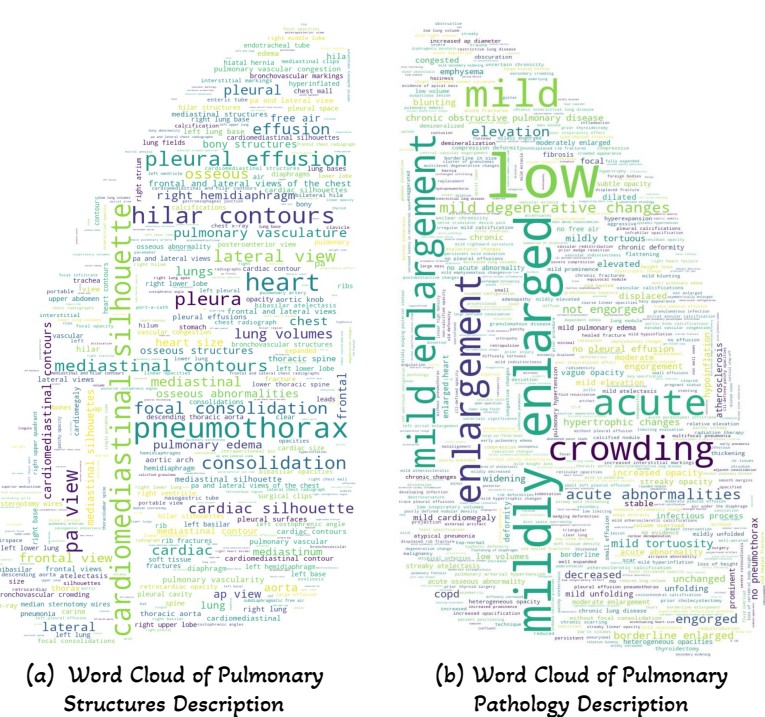

(a) Word Cloud of Pulmonary Structures Description

(b) Word Cloud of Pulmonary Pathology Description

Figure 1: The word cloud illustrating descriptions of pulmonary structures and pathologies.

**Category of Annotated Visual Cues.** The visual cues in CoVT-CXR annotations encompass 112 categories, covering five annotation types: bounding box, mask, point, line, and linestrip. The details of these categories can be found at here. We further demonstrate the number of mask annotations in Fig. 2 where the original 112 categories are grouped into 21 classes for visualization purposes.

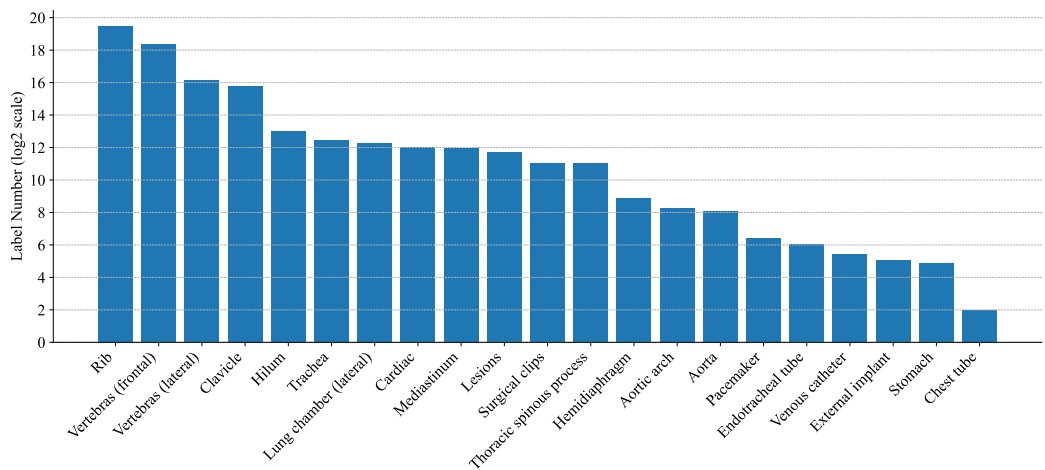

Figure 2: Number of mask annotations in $\mathcal{T}$ based on 21 high-level categories.

**Rationale Step Statistics.** As described in our main paper, each diagnostic report $\mathcal{R}$ can be divided into multiple sentence segments $S$. By definition, each $S$ corresponds to one field, referred to as the target field. Our CoVT-CXR dataset explicitly analyzes a single chain-of-thought reasoning process in one target field. As can be found in Fig. 3, we have at most 12 steps of reasoning for each segment $S$, indicating the complexity of the reasoning process. The projection tag, which describes the type of imaging views, requires only one step of reasoning. This is reasonable because frontal imaging with AP and PA views, as well as lateral imaging, can be directly defined by the given CXR images. The bone field descriptions exhibit a relatively uniform distribution of step lengths ranging from 1 to 11, likely due to the variability in the types and quantities of bones being examined. Reports related to the lung field show either short or very long step lengths, possibly due to the presence of simple (single nodule) or complex (multiple abnormalities) diagnoses. Cardiac-related reports tend to cluster around intermediate step lengths of 7-8, reflecting the predetermined heart measurement protocols, which typically require 6 steps to complete the cardiothoracic ratio calculation. This analysis highlights the nuanced relationship between diagnostic complexity and reasoning processes, underscoring the importance of tailored approaches in medical image interpretation. We hope that this observation can guide the development of more efficient and accurate diagnostic tools in the future.

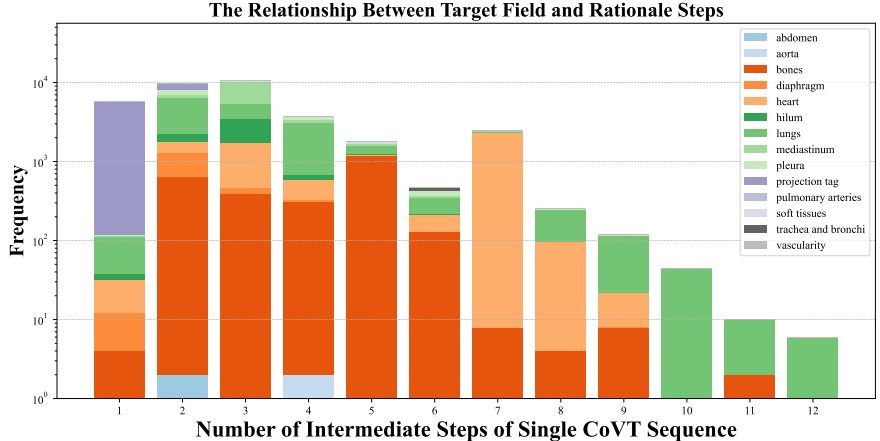

Figure 3: Statistics on number of reasoning steps in one segment sentence $S$ in CoVT-CXR dataset.

**Ribs Distribution Notation.** Some may notice a chunk of ribs or vertebrae presented in the sunburst figure in the main paper. However, reports of bone lesions are relatively scarce in the MIMIC-CXR (Johnson et al., 2019). Although rib lesions account for only a small portion of the

MIMIC-CXR dataset, ribs play a crucial role in intermediate reasoning steps, thereby are significant in our metadata. For instance, ribs help localize lesions and indicate signs of pulmonary collapse, hyperinflation, or atelectasis, as they are consistently present and orderly arranged within the lungs. These abnormalities, frequently reported by MIMIC-CXR, require precise identification of ribs. Statistically, we analyzed the frequency and percentage of occurrences of these pathologies in 120k findings extracted from the official dataset. Results in Tab. 1 indicate that rib involvement in related lesion descriptions spans a wide range, with both direct and indirect participation in diagnoses exceeding 40%.

| Class | Ribs | Collapse | Hyperinflation | Atelectasis | Involvement of ribs |
|---|---|---|---|---|---|
| Count | 10133 | 15456 | 4611 | 29485 | 48117 |
| Percentage | 8.44% | 12.88% | 3.84% | 24.57% | 40.10% |

Table 1: The count and proportion of diagnoses involving the ribs in the raw dataset.

## B    ANNOTATION PHASE

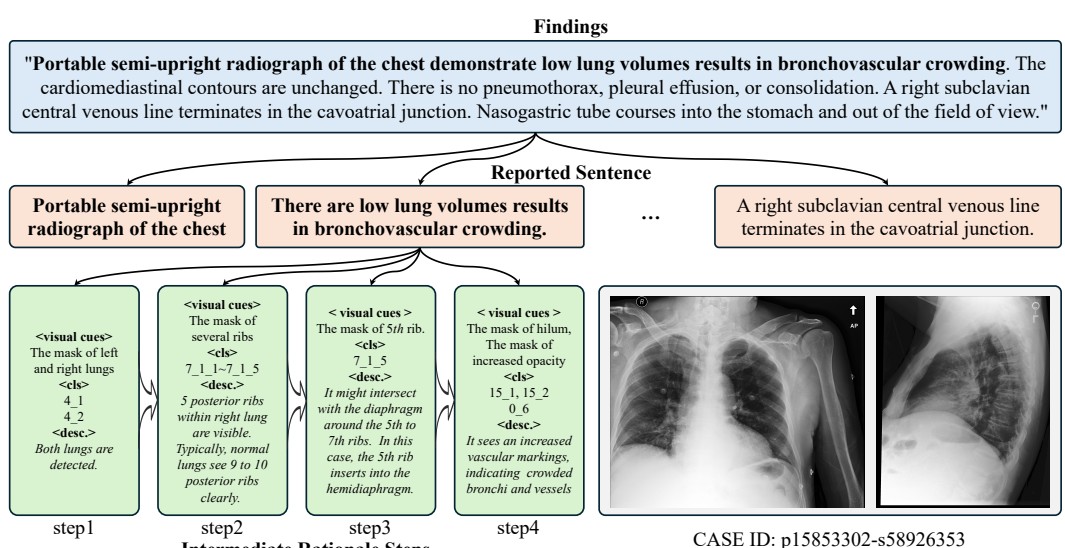

Figure 4: Illustration of five tasks derived from CoVT-CXR, which align our five steps well.

**Obtaining Metadata.** In Fig. 4, we provide an example that indicate how medical trainees collaborate to obtain metadata $\mathcal{T}$ from $\{I, \mathcal{R}\}$. Our $\mathcal{R}$ is represented as "Findings," with CXR images shown in the bottom-right of Fig. 4. Specifically, $\mathcal{R}$ is broken down into multiple $S$ segments. We further emphasize our semantic disentanglement by bolding sentences where a single sentence in $\mathcal{R}$ can be divided into two $S$ segments. For each sentence $S$, medical trainees are required to provide four different $\mathcal{T}$ corresponding to various stages derived from the ultimate task **T5**. For example, step 1 is the segmentation task where annotators provide visual cues $v$, or masks in this case, for the class $c$, such as 4-1 and 4-2 for the left and right lungs, along with the description $d$, "*Both lungs are detected*".

**Annotation Tool Features.** To collect $\mathcal{T}$ from $I, \mathcal{R}$, we adapted Labelme (Wada) to develop our annotation tool, Labelme-CoVT, incorporating the following key features:

- *Textual Component Integration:* This establishes connections between the CXR image $I$ and the report $\mathcal{R}$.

- *Flag Module Modification:* This serves as a reference for reported sentences $S$, allowing for the disentanglement or merging of sentences.

- *SAM Integration for CXR:* The SAM-CXR model, trained for **T1**, assists in expediting the labeling process. This feature allows users to interactively create masks through simple

clicks or points. Additionally, SAM for CXR within the annotation tool can accept labelers' modifications as human feedback for further incremental learning.

More annotation features can be seen in Labelme-CoVT demo.

**Annotation Simplify and Efficiency Improvement.** Given the intricate rationale workflow, the Labelme-CoVT tool plays a pivotal role in curating the CoVT-CXR dataset. It not only facilitates the visualization of sequential interleaved cross-modal data for simplification but also accelerates the labor-intensive manual annotation process by streamlining verification and improving efficiency. We randomly selected four groups from a pool of 32 medical trainees, who were required to complete a questionnaire assessing time consumption. Additionally, we measured inter-annotator agreement among the trainees to validate the completed cases. Tab.2 summarizes the efficiency and accuracy outcomes of the annotation phase.

| Group | #1 | #2 | #3 | #4 |
|---|---|---|---|---|
| Avg. Time (w/o tool) | 43 | 40 | 40 | 45 |
| Avg. Time | 14 | 8 | 9 | 12 |
| Speed up | **3.1x** | **5x** | **4.4x** | **3.8x** |
| Cross Person Agreement (w/o tool) | 62.12% | 45.12% | 50.44% | 75.41% |
| Cross Person Agreement | 85.12% | 91.23% | 87.33% | 90.53% |
| Agreement Improvement | **+23.00%** | **+46.11%** | **+36.89%** | **+15.12%** |

Table 2: Evaluation of Labelme-CoVT in annotation simplification and efficiency improvement.

The time consumption, measured in minutes, shows that the tool significantly increases annotation efficiency, reducing the time required by three to five times. Furthermore, it enhances annotation accuracy by approximately 1.5 times, with data usability surpassing 80%.

**Data Assignment for Multi-stage Training.**

We follow a progressive training scheme to gradually improve the model's multi-step reasoning capabilities. For instance, we incorporate single-step reasoning tasks like segmentation or image description during pretraining, and further fine-tune the model with chain-of-visual-thought reasoning using long contextual data. As a result, our multi-stage training setup and data volumes are arranged as shown in Tab.3.

| Stage | Task | Format | Volume |
|---|---|---|---|
| Pretrain | T1 | $\langle I, p, v' \rangle \rightarrow v$ | 0.4M |
| | T2 | $\langle I, p, v \rangle \rightarrow t$ | 2.4M |
| | T3 | $\langle I, q \rangle \rightarrow \langle v_1, \ldots, v_i \rangle$ | 0.2M |
| Finetune | T4 | $\langle I, q' \rangle \rightarrow \langle (v_1, t_1), \ldots, (v_i, t_i), S \rangle$ | 30k |
| | T5 | $I \rightarrow \{\langle \{v_{ij}^m, t_{ij}^m\}_{i_j=1}^{i_m}, S^m \rangle\}_{m=1}^M$ | 6k |

Table 3: The count and proportion of diagnoses involving the ribs in the raw dataset.

Notably, there are more entries for **T2** than the others. This is due to our design, where versatile $p$ and $v$ of specific image $I$ are randomly combined to enhance the robustness of text generation.

**Separating the SAM-CXR and CoVT Pretraining Data.** The **T1** task data is used both to assist the build SAM-CXR and CoVT pretrain after the annotation process. Initially, one-third of the data is annotated manually, and this data is then used to train the SAM-CXR for CoVT-CXR data curation. The trained SAM-CXR is subsequently employed to assist in annotating the remaining two-thirds of the data. Once the annotation process is complete, the fully-annotated dataset is then used to update all relevant models, including SAM-CXR and CoVT.

## C  GPT PROMPT FOR CONTEXTUAL REFINEMENT & GENERATION

```
# Define the intermediate step flow
intermediate_step_tmp = dict()
for step_id, step in enumerate(intermediate_step):
        intermediate_step_tmp[f"rationale of step{step_id + 1}"] = \
            {"type": "string", "description": f"the rationale description of step{step_id + 1}"}
intermediate_step_des = str()
for step_id, des in enumerate(intermediate_step):
        intermediate_step_des += f"step {step_id + 1}: {des},\n"
```
---
```
# Define the tools
Function Name: Intermediate diagnostic rationale generation
Function Description:
        "There is one or several highlighted image-text pair showing intermediate steps with a final diagnostic report for
        interpreting a chest X-ray. Based on this report, please provide the finding clues and rationale of diagnosing this
        report for every mentioned step in this case. Do not mention the given specific step number and provided raw
        description, and retain the numerical content appearing in each step. The current rationale could be able to
        reference the previous images and generated rationales, and should gradually lead towards the final report. The
        given step input and the generated rationale of step must be one-to-one."
Parameters:{
        "type": "object",
        "properties": intermediate_step_tmp,
        "required": list(intermediate_step_tmp.keys())
}
```
---
```
# Define the messages
messages = [
        {"role": "system", "content": f"This is auxiliary information: Request ID: {request_id}"},
        {"role": "user", "content":
                The final report description states that these findings suggest that "{final_reported_sentence}".
                The intermediate step descriptions are as following: \n{intermediate_step_des}
        },
]
request_list.append(gpt_request_wrap(model_name, messages, tools))
```

Table 4: Rationale refinement for $d$.

**Rationale Refinement.** Unlike machines, human descriptions $\mathcal{R}$ can vary in levels of abstraction and may omit background information, considering the conversational context during communication. While this approach is efficient among doctors, it is machine-unfriendly, particularly when deep reasoning is required based on the provided descriptions. To this end, we propose incorporating external knowledge, such as LLMs, to augment chain-of-thought reasoning descriptions. In practice, we use the GPT-4 (OpenAI et al., 2024) API to refine descriptive content, as illustrated in Tab. 4. Specifically, we first convert the class $c$ to a textual description with the phrase, "the highlighted area showing the <c>". We then input this converted description, the raw textual descriptions provided by annotators at the current reasoning step, and the corresponding sentence segment $S$ into the LLMs. The LLMs can understand the causal relationship between the annotated content at the current diagnostic step and the formation of the given final diagnostic sentence. For example, when we provide the annotation of five ribs along with a statement indicating reduced lung capacity, GPT-4 may generate a rationale like, "Only five ribs are visible; normally, 9-10 ribs can be seen with fully expanded lungs." This sentence becomes the final textual description $d$ at the current reasoning step. This design helps in forming a more reliable chain-of-thought process, leading to a more confident diagnostic reasoning process by leveraging the strengths of LLMs to support medical professionals in making informed decisions.

**Instruction Generation.** As stated in our main paper, **T4** requires instructions $q'$, or equivalently questions, as input, and GPT-4 is utilized to generate these instructions. However, simply applying GPT-4 on whatever sentence $S$ would lead to overly specific or vague questions as sentences from pathological or normal reports vary a lot. This discrepancy can mislead the CoVT model by causing it to focus on the questions rather than considering the CXR images. For instance, sentences from a pathological report typically offer detailed descriptions of a lesion, leading to highly specific questions about that lesion. Conversely, sentences from a normal report are often oversimplified, causing GPT-4

```
# Define the tools
Function Name: Medical question generator
Function Description:
        Given a report sentence of a CXR image, complete the content according to the following requirements.
        1) Please complete it into a full report sentence without changing the meaning, Delete blank descriptions and any
           descriptions related to comparisons with previous results.
        2) After that, decide which field the given sentence refers to.
        3) Arrange a question that can be confidently answered by the reported sentence.
        Never generate a too-specific question that indicates the position or clues of disease directly. If there any devices,
        ask what findings/condition in related field"
Parameters:{
        "properties": {
        "completed_sentence": {
                "type": "string",
                "description": "The completed report sentence."
        },
        "field": {
                "type": "string",
                "description": "The field that the sentence refers to.",
                "enum": ["lungs", "pleura", "mediastinum", "heart", "bones", "diaphragm", "trachea and bronchi", "soft
                tissues", "hilum", "projection tag"]
        },
        "generated_question": {
                "type": "string",
                "description": "The course question to ask the filed's findings or condition. "
        },
        "required": [ "completed_sentence", "field", "generated_question"],
}

# Define the messages
for request_id, reported_sentence in enumerate(samples):
        messages = [
                {"role": "system", "content": f"This is auxiliary information: Request ID: {request_id}"},
                {"role": "user", "content": reported_sentence}
        ]
        request_list.append(gpt_request_wrap(model_name, messages, tools))
```

Table 5: Generated question instruction $q'$ for visual question answering.

to generate vague questions more frequently. To avoid this, GPT is asked to identify the relevant field for each report sentence first. Then it formulates questions based on this field, as demonstrated in Tab. 5. During inference, our CoVT takes $\{I, q'\}$ as input, generates a chain-of-thought process $\langle (v_1, t_1), \ldots, (v_i, t_i) \rangle$, and finally provides sentence $S$.

# D   IMPLEMENTATION DETAILS

**Implementation of SAM for CXR (T1).** We apply the SAM (Kirillov et al., 2023) model for CXR segmentation, choosing the smallest configuration with 12 vision encoder layers and 768 visual embedding channels, along with the default mask decoder setup. To adapt SAM for CXR segmentation, we modify the prompt encoder by removing the bounding box prompt due to the irregular pulmonary structures and using a point sample strategy within 5 iterations. Additionally, we introduce 76 category prompts to compensate for the absence of a text prompt encoder, as SAM reportedly uses CLIP (Radford et al., 2021). The model is trained in mixed precision with the AdamW optimizer (Loshchilov & Hutter, 2019), using a learning rate of 1e-3 and a weight decay of 0.1. Gradient accumulation is employed to increase the batch size to 128. The training process of SAM for CXR takes more than 48 hours with 12 RTX 4090 GPUs.

**Implementation of VQGAN for Tokenization.** We set the autoencoder (Esser et al., 2021) to have a discrete codebook size of 8192 with a latent dimension of 256. Given a resized image of resolution 336, the image encoder reduces the resolution to 21, resulting in an image or mask taking a contextual length of $21^2 = 441$ in the subsequent foundation model. The model employs L1 reconstruction loss and LPIPS perceptual loss (Zhang et al., 2018). We train the tokenizer on 1.4M samples of

various masks and image combinations from the CoVT-CXR dataset for 250k iterations. Generative adversarial loss is enabled at the $10,000th$ step when the reconstructed image achieves a certain level of quality. We set the batch size to 48 and it takes us 30 hours on 12 RTX 4090 GPUs to complete the tokenizer training process.

**Implementation of Foundation Model for CoVT.** We adopt Llama3-Instruct-7B (Meta, 2024) as our base model and redefine the model's embedding layer to map visual tokens from VQGAN and textual tokens into a unified embedding space, encompassing representations for both modalities. Subsequently, we perform full-parameter pretraining on a dataset of 1.3 million extracted samples. During the pretraining phase, we use 8 H100 GPUs with a batch size of 128 and a learning rate of 1e-4, taking 24 hours to complete one epoch. In the SFT phase, we fine-tune the model on the CoVT-CXR dataset with a batch size of 64 and a learning rate of 4e-5, training for 4 epochs in 3 hours.

# E    MORE DETAILS ABOUT EXPERIMENTAL RESULTS

**T1: SAM-CXR.** As a reminder, **T1** is a **image-to-image** task aiming to mimic the recognition process of doctors, which is fulfilled with SAM framework. In experiments, we take Meta SAM (Kirillov et al., 2023) and MedSAM (Ma et al., 2024) as our baselines; the former performs strong zero-shot capabilities on various natural images, while the latter has undergone extensive fine-tuning for 2D medical images. Given the significant variability in image distributions of different structures in CXR, we choose five of them, namely the trachea, lungs, heart, bones, and mediastinum, to represent the majority of visible structures in CXR. We exploit point-wise prompts for SAM and SAM-CXR for its simplicity and effectiveness.

| Model | Trachea | Heart | Lungs | Ribs | Clavicles | Mediastinum | Avg. |
|---|---|---|---|---|---|---|---|
| SAM(Kirillov et al., 2023) | 3.89 | 23.66 | 78.79 | 5.23 | 9.08 | 24.91 | 24.26 |
| MedSAM(Ma et al., 2024) | 49.42 | 75.01 | 89.70 | 15.82 | 10.87 | 70.23 | 51.85 |
| CXR-SAM | **74.11** | **86.88** | **98.41** | **84.70** | **91.70** | **88.16** | **90.76** |

Table 6: The comparison of Meta SAM, medical image fine-tuned MedSAM, and SAM-CXR in CXR applications. The evaluation metric used is Intersection over Union (IoU), with 'Avg.' refering to the average IoU score when considering all reported structures

In contrast, MedSAM is trained exclusively with a bounding box (bbox) prompt encoder thus bbox-wise prompts are used in our experiment. Results in Tab. 6 show that the general SAM model performs reasonably well in lung segmentation, likely due to the distinct segmentation boundaries of the lungs. However, it fails in segmenting the heart, mediastinum, and bones. We attribute this to the overlapping structures in X-ray imaging, which obscure the target image features. In addition, it is worth noting that the MedSAM fine-tuned on medical images seems to perform better than SAM. This might be because the bbox-wise prompts, which are used by MedSAM, provide more precise location information compared to point-wise ones.

Again, the low accuracy in segmenting the most challenging bones suggests that MedSAM also fails on CXR segmentation. Our approach successfully addresses the shortcomings of the SAM model in CXR segmentation, providing a robust visual foundation model for downstream CXR tasks.

| Metric | BLEU-1 | BLEU-2 | BLEU-3 | BLEU-4 | ROUGE-1 | ROUGE-L | METEOR | CIDEr |
|---|---|---|---|---|---|---|---|---|
| Value | 0.3877 | 0.2404 | 0.1563 | 0.1020 | 0.5154 | 0.3328 | 0.3221 | 0.3529 |

Table 7: Benchmark performance for the **T2** task

**T2: Imaging Caption.** The design of **T2** is inspired by the process where doctors proceed to draw conclusions or summarize results in **T1**, thereby is a **image-to-text** task.

Similar to (Huang et al., 2023), our CoVT focuses on summarizing a prompted region rather than describing the entire CXR image, which turns out to be beneficial for understanding finer details in CXR images.

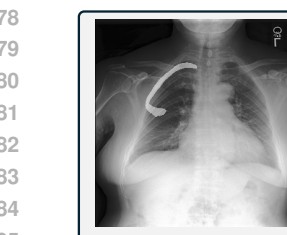

**Q:** What does the mask indicate in this CXR image?

**GT:** The highlighted area in the CXR image indicates the location of the right 3rd rib as observed on a frontal view. The reference to 'the mask of right 3rd rib' likely means that this specific rib has been marked or identified for some reason, possibly due to an abnormality or area of interest that the radiologist needs to examine closely. This could be for an evaluation of fractures, lesions, or any other pathology localized to this rib. Further details from the image itself would be necessary to provide more specific insights.

**CoVT:** The highlighted segment in the CXR image corresponds to the anatomical location of the right 3rd rib as seen on a frontal view. This area represents the intersection of the rib with the X-ray beam, which can appear as a dense, linear shadow. There are no abnormalities such as fractures, lesions, or abnormal bone density.

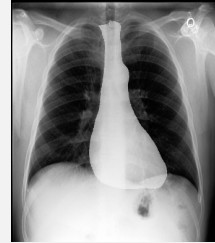

**Q:** What feature is highlighted in this CXR image?

**GT:** The highlighted area in the X-ray indicates the cardiacmediastinal or mediastinal region on the frontal view. This area often pertains to the region encompassing the heart and mediastinum. Abnormalities in this region may suggest a range of conditions such as cardiomegaly (enlarged heart), mediastinal mass (e.g., lymphadenopathy, tumors), or other anomalies like an aortic aneurysm or pericardial effusion. Interpretation would depend on the specific characteristics of the highlighted area, such as its size, shape, and density, as well as accompanying clinical information and symptoms.

**CoVT:** The highlighted area in the chest X-ray indicates the cardiacmediastinal or mediastinal region, which is an area of interest in the central part of the chest. This region includes the heart, major blood vessels like the aorta and pulmonary arteries, as well as other structures. Abnormalities in this area can suggest a variety of conditions, such as an enlarged heart (cardiomegaly), mediastinal mass, lymphadenopathy, or fluid accumulation (mediastinal effusion). A detailed evaluation is needed to provide a specific diagnosis, often requiring correlation with clinical findings and possibly further imaging or tests.

Table 8: Exsamples for visual question answering. The model highlights specific regions based on the question and further generates region-specific relevant descriptions based on the content of the highlighted areas.

Fig. 8 and Tab. 7 show the qualitative and quantitative results of **T2** on CoVT-CXR respectively. Furthermore, thanks to SAM-CXR, we can interactively generate captions for specific regions in CXR images using points, masks, or text prompts. Please check the demo for the interactive caption generation process.

| | |
|---|---|
| ➢Locate the carina of the bronchus | ➢Detect the bottom contours of the lateral lung cavity |
| ➢Locate the contours of the left and right heart | ➢Detect the blunted costophrenic angle in the lateral view |
| ➢Setting board line of cardiac contours | ➢Frontal view right rib blunted contour |
| ➢Measuring the width of heart/cardia | ➢Frontal view left rib blunted contour |
| ➢Detect abnormal heart contours | ➢Find the tip of the ET tube |
| ➢Locate the board line of the outer contours of both lungs | ➢Measure the distance between the tip of ET tube and the carina |
| ➢Measure the width of both lungs | |
| ➢Detect the contour of the aortic arch | ➢Check if the NG tube is in the stomach |
| ➢Detect mediastinal contours | ➢Detect the pacemaker with electrode |
| ➢Find abnormal protruding mediastinal contours | ➢Find the tip of the pacemaker electrode |
| ➢Trace the vertebral line of the thoracic spine | ➢Find the end of the venous tube |
| ➢Detect the hilum contours | ➢Locate the tip of the chest tube or drainage tube |

Table 9: Predefined instruction for building visual reasoning chain.

**T3: Instruction-following Visual Thought.** Apart from the cross-modal chain of visual thought, we recognize that single visual reasoning is also beneficial for accomplishing clinical-related tasks, such as coarse-to-fine detection, marking subtle points, and performing complex medical measurements. Tab. 9 displays our predefined instructions used to guide single-step visual reasoning. The specific meanings of the instructions can be found in here, where class id mentioned are following categories of A. It is worth noting that predefined instructions can be combined to form a series of visual reasoning steps to complete more complex clinical-related tasks. More details for visual reasoning guided by instructions can be seen in visual reasoning pipeline.

**Why is the multi-step visual thought design helpful?** To explicitly showcase the impact of multi-step reasoning, we provide case-wise explorations on the average sampling probability on the ground truth (GT) for final findings across varying numbers of visual cues. As evidenced in Fig. 5, incorporating additional intermediate visual cues and translating them into meaningful sentences significantly increases the average sampling probability of GT tokens, thereby improving the likelihood of generating accurate GT reports.

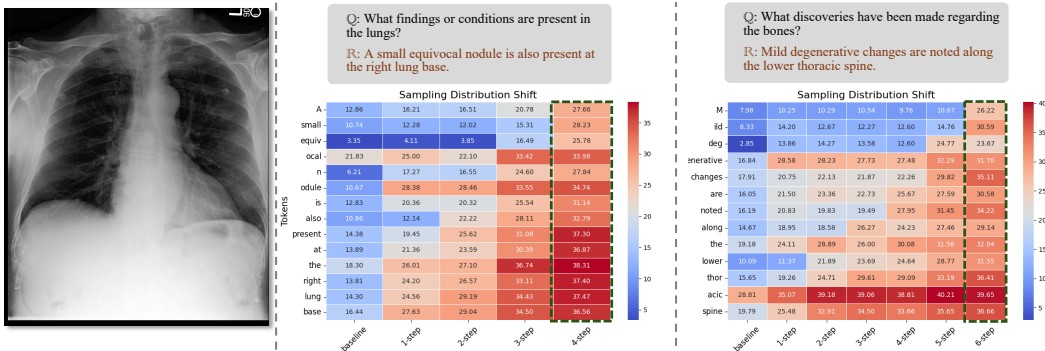

Figure 5: Ground truth sampling gains in token-level as intermediate steps increases.

## F    VISUALIZATION OF CoVT RESULTS

**Generalization for Unseen Data.** CXR images, unlike CT or MRI, exhibit a more consistent distribution, distinguishing them from other medical imaging. Consequently, SAM-CXR and CoVT have shown strong generalization capabilities on unseen data, as illustrated in Fig.6. SAM-CXR excels in segmenting bones and airways via prompt-based segmentation, which are typically challenging to discern by Meta SAM (Kirillov et al., 2023) and medical SAM (Ma et al., 2024) due to overlapping shadows in radiographs. CoVT also performs well on datasets like Vindr (Nguyen et al., 2022), being less affected by distribution shifts. This is due to the precise visual prompts in Chain-of-Visual-Thought, which reduce noise by focusing on highlighted bone pathology regions, avoiding interference from surrounding tissues. Additionally, CoVT aids in identifying CXR views and lesions, enabling interpretable reasoning process through single or multi-step diagnostic tasks.

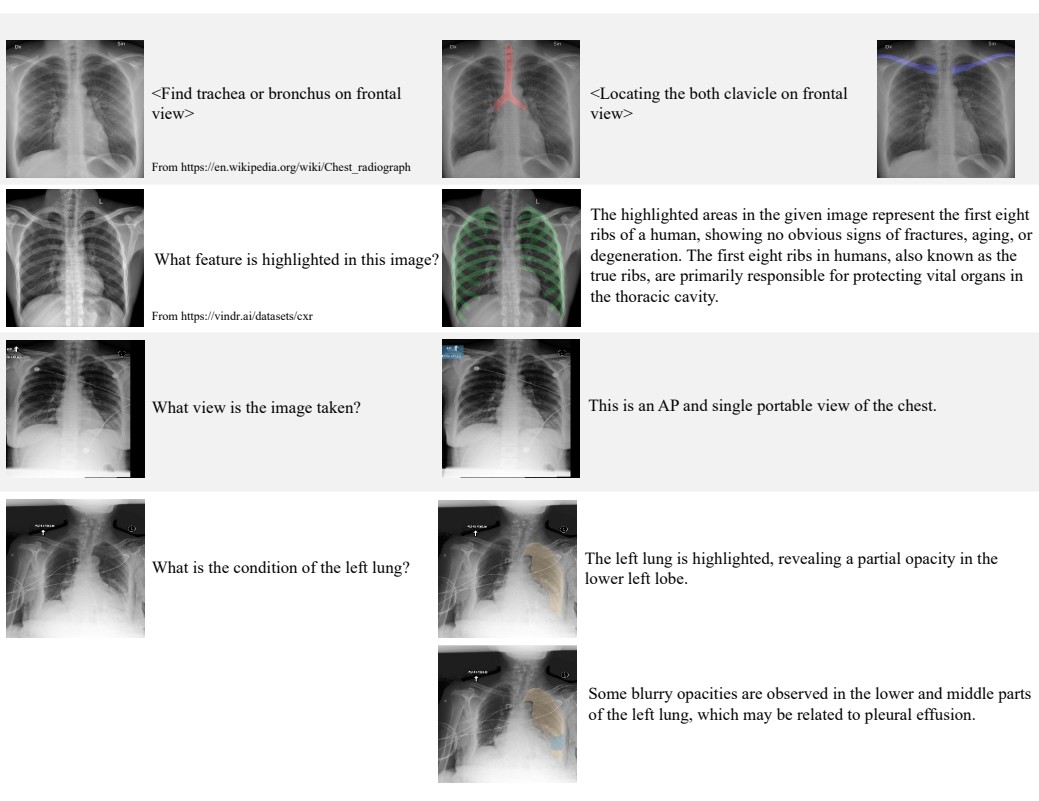

Figure 6: Generalization test visualization from several unseen data.

**Chain of Visual Thought Visualization.** We provide some examples of our CoVT results on CoVT-CXR on task **T4** and **T5** in Fig. 7 and Fig. 8 respectively. Clearly, our CoVT generates more accurate answers and reports compared to existing methods, demonstrating its superiority over the baselines. More importantly, CoVT provides interpretable cross-modal intermediate reasoning steps, enhancing the reliability and interpretability of state-of-the-art methods.

**Q:** What is the projection view of the chest?

**GT:** AP view of the chest

**Gemini-1.5-Pro:** The projection of this chest X-ray is anteroposterior (AP). This means the X-ray beam passed from the front (anterior) of the patient to the back (posterior). You can tell this is an AP view because Scapulae are not cleared: The scapulae (shoulder blades) are superimposed over the lung fields, which is typical of an AP projection where the patient can't always pull their shoulders forward effectively.

**GPT-4V:** The projection view of the chest X-ray provided is an anteroposterior (AP) view, typically taken in the upright position, as indicated by the labels "PORTABLE" and "UPRIGHT ".

**CoVT:** *The bounding box of the word "portble" in image indicates an AP view of the chest.*

*<findings>Single AP view of the chest</findings>.*

**Q:** What are the findings of the mediastinal silhouettes?

**GT:** Calcification of the aortic knob is seen.

**Gemini-1.5-Pro:** The mediastinal contours are within normal limits. There is no mediastinal widening.

**GPT-4V:** The mediastinal contours appear normal with no signs of widening.

**CoVT:** *The mediastinal structures appear to be normal.* *The aortic arch is not prominent.* *There is a calcification of the aortic*

*<findings>calcification of the aortic knob is seen.</findings>*

**Q:** What is the condition of the lungs?

**Gemini-1.5-Pro:** The lungs are clear bilaterally. There is no evidence of consolidation, pleural effusion, or pneumothorax.

**GPT-4V:** Based on the provided chest X-ray image, the lungs appear clear without signs of consolidation, effusion, or pneumothorax. There are no obvious abnormalities such as masses or lesions observed within the lung fields. The costophrenic angles are sharp, indicating no pleural effusion. Overall, the lungs seem to be in a normal condition.

**CoVT:** *the right and left lung areas are examined on the frontal view. both lungs exhibit clear fields without any signs of focal consolidation. additionally, there are no visible signs of pleural effusion or pneumothorax, which supports the final report*

*<findings>There is no focal consolidation, pleural effusion, or pneumothorax identified.</findings>*

Figure 7: One example of our CoVT for VQA task based on **T4**

**Report:** The lungs are clear of consolidation, effusion, or pneumothorax. The cardiac silhouette is top normal in size. Mildly tortuous descending thoracic aorta is seen. No acute osseous abnormality detected.

**User:** Can you assist with writing a diagnostic report?

**Gemini-1.5-Pro:** Lateral chest radiograph demonstrates clear lungs without focal consolidation, pleural effusion, or pneumothorax. The cardiomediastinal and hilar contours are within normal limits. No acute osseous abnormalities are seen.

**GPT-4V:** The chest X-ray images reveal clear lung fields with no evidence of focal consolidations, pneumothorax, or pleural effusions. The cardiac silhouette appears normal in size, and the mediastinal contours are within normal limits. The bony thorax, including the ribs, clavicles, and spine, show no obvious abnormalities or fractures. The diaphragms are well-positioned and without abnormal elevation or flattening. Overall, the chest radiographs are unremarkable, with no significant pathology detected.

**CoVT:**

*PA and lateral views of the chest obtained.*
*<findings>PA and lateral views of the chest</findings>.*

*The lungs appear to clear without consolidation, effusion, or pneumothorax*

*The lateral view also shows no obvious abnormalities.*

*There is no blunting of the costophrenic angles in the posterior lung bases, suggesting the absence of pleural effusion.*

*<findings>Clear lungs with no consolidation, effusion, or pneumothorax. </findings>*

*The cardiac silhouette is stable.*

*Measure the heart width.*

*Identify the structures of both lungs.*

*Measure the lung width; the cardiothoracic ratio is less than 50%, indicating no cardiomegaly.*

*<findings>The heart size is within normal. </findings>*

*The mediastinal structures are normal.*

*The descending aorta tends to be mildly tortuous.*

*<findings>The mediastinum is unremarkable, but there is mildly tortuous descending thoracic aorta </findings>*

*The clavicle is intact.*

*The ribs show no obvious abnormalities.*

*The thoracic spine is located at the midline.*

*The lateral view shows no significant degeneration of the vertebrae.*

*<findings>No acute osseous abnormality. </findings>*

Figure 8: One example of our CoVT for report generation task based on **T5**