# OpenReview forum: "CoVT-CXR: Building Chain of Visual Thought for Interpretable Chest X-Ray Diagnosis"
_ICLR.cc/2025/Conference — Submitted to ICLR 2025_

### Official Review · Reviewer_fMuk · 2024-10-18

**Soundness:** 3
**Presentation:** 2
**Contribution:** 3
**Rating:** 8
**Confidence:** 3

**Summary:**

In this work, the authors propose a chain-of-thought approach to automated radiology report generation. They propose a data labeling strategy and tool to support this approach, generate a new set of chain-of-thought labels for a subset of the MIMIC-CXR dataset, and propose a new training strategy to utilize their chain-of-thought labels.

**Strengths:**

-	A new dataset has been developed that should be valuable to the community. This dataset contains fine-grained, chain-of-thought reasoning labels that mimic how radiology reports are generated by humans. This dataset generation is a manually intensive process that will likely support future research.
-	An open-source tool to generate additional chain-of-thought labels was developed and provided, which is another valuable addition to the community.
-	In addition to generating more correct answers, the proposed method also generates more interpretable answers, which is a critical need in this domain.

**Weaknesses:**

-	The authors reference an Appendix many times, but there is no Appendix.
-	The baselines in the evaluation are not medical-specific models. There have been many models proposed for radiology report generation which have been trained on medical images/text, however the authors opt to use natural image/text baselines, which is an unfair comparison. For example, instead of comparing against Llava-med or llava-rad, they compare against llava; instead of comparing against med-gemini, they compare against gemini. Similarly, they exclude models specifically trained for radiology report generation, such as maira-1 or any of the numerous other proposed radiology report generators. As a result, I do not find the empirical results convincing.
-	The paper is difficult to read and follow. In particular, I have a difficult time following how the dataset was created and the contents of the dataset. What are the 112 semantic classes? How were these chosen? What is a “intermediate text description”? Is there any standardization on what an “intermediate text description” comprises? The murkiness of this dataset development description makes it difficult to understand how valuable the dataset will be to others or how one would go about extending the dataset.
-	Additionally, the total size of the dataset is relatively small (6k cases), and the test dataset is very small (400 cases). While I understand this is because of the manual effort required to create a chain-of-thought dataset, it does limit its upper bound on performance. I am doubtful of how well this dataset size could be increased in the future, as the chain-of-thought labelling process is not very well described. Further, if the authors could discuss how this dataset could be used in tandem with other non-chain-of-thought datasets, that would be useful.
-	In the Data Curation: Motivation section and later in Section 3.1, the authors authoritatively describe a process that doctors (the authors imply all doctors) use to read chest x-rays. There are no references in this section, aside from a footnote pointing to “geekymedics” and “radiologymasterclass” websites. I find this support weak; either the language in this section should be made less strong, or additional support is needed as to how the authors determined this CXR reading process.
-	Limitations and Future Work should be discussed.

In summary, I find the dataset and approach interesting and useful for the community. While the baseline comparisons leave me unsure of how well the proposed method actually works and the writing needs revision for clarity, I still find this dataset and general approach to be a valuable contribution and lean toward acceptance.

**Questions:**

Above.

---

### Official Review · Reviewer_Hpdb · 2024-11-02

**Soundness:** 3
**Presentation:** 4
**Contribution:** 3
**Rating:** 6
**Confidence:** 4

**Summary:**

This is a dataset and benchmark paper that incorporates the idea of a 'chain of thought' into constructing the Chest X-ray dataset. This paper offers a unique insight; instead of training end-to-end medical VLM models as most works tend to do, it proposes that just like clinicians make diagnoses based on observations and changes in thought, this process should be incorporated into the dataset. Overall, I believe this paper could be helpful to many audiences. However, there are two issues with this paper: more standard models should be used to compare with CoVT, and using only general evaluation metrics is not sufficient for medical report generation for VQA. I initially gave the paper a weak acceptance, but I might change my score based on whether the paper can provide more evaluation results, especially on medically factually related accuracy scores.

**Strengths:**

1. *Novel Insight*: It offers a unique insight by proposing that just like clinicians make diagnoses based on observations and changes in thought, this process should be integrated into the dataset.

2. *Comprehensiveness of the Dataset*: The dataset is extensively annotated by over 30 annotators. The medical questions cover a wide range of diseases, enhancing its applicability and relevance.

3. *Claimed Transparency*: Although not available yet, the authors commit to releasing both the model and the annotation tool, along with the related codes. This promised transparency is commendable and aligns with open science principles.

**Weaknesses:**

1. *More model evaluation*: The use of LLaVA-1.5 is noted, but LLaVA-1.6 offers significant improvements that could enhance model performance. I recommend upgrading to this latest version for fine-tuning. Additionally, exploring models like Flamingo may further improve results.
2. *Evaluation Metrics*: The evaluation metrics used—BLEU, ROUGE, METEOR, and CIDEr—are standard for many natural language processing tasks; however, they are not ideally suited for evaluating medical Visual Question Answering (VQA) or the factual accuracy in medical reports. These metrics often fail to capture the semantic accuracy required in medical contexts. It is critical to implement more specialized metrics that can assess the medical factual accuracy and the clinical relevance of the answers provided by the system.

**Questions:**

1. *Model Comparison*: Do you think it is a fair comparison to have an encoder trained on medical data while a model like LLaVA uses an encoder trained on everyday images?

2. *Image encoder*: The SSL method (VAE) you used is somewhat outdated. Are there specific reasons for not using more up-to-date methods like Dino_v2? I'm not suggesting that what you did is incorrect or insufficient; I just want to understand why you chose this particular SSL method.

3. *Multi-Stage Fine Tuning*: The approach you took to multi-stage fine tuning is interesting. Do you have any evidence supporting that what you did is necessary?

---

### Official Review · Reviewer_Rqi3 · 2024-11-03

**Soundness:** 3
**Presentation:** 3
**Contribution:** 3
**Rating:** 8
**Confidence:** 5

**Summary:**

This work contributes a new public dataset of chest X-ray interpretation (and the chain of thought used) designed for use training Visual LLMs to perform radiology tasks. A model is trained with this data and performance is evaluated against current visual LLMs.

**Strengths:**

This work presents improved performance compared to zero shot of existing VLM models.

The ablation studies are well done. They present evidence demonstrating multi step reasoning improves model performance compared to a step for medical analysis tasks.

**Weaknesses:**

The dataset appears through, designed by clinical experts, and cast as well studied ML tasks such as segmentation and question answering.

The evaluation metrics used are generally for machine translation and don't capture how useful the reports are, only that they contain similar words. A solution would be a human evaluation for some set of tasks with quantitative feedback.

**Questions:**

It would be nice to have more insight into the topics, pathologies, and tasks included in the dataset. Such as a table showing these categories and the counts of samples associated with them. This would help to identify what is in-domain for the model.

---

### Official Review · Reviewer_yTXu · 2024-11-04

**Soundness:** 2
**Presentation:** 2
**Contribution:** 2
**Rating:** 5
**Confidence:** 4

**Summary:**

This work introduces a multi-step instruction tuning framework for generating radiology reports and performing visual question answering (VQA) for chest radiographs (CXRs). To train and fine-tune their models, they also introduce an annotated subset of the MIMIC-CXR dataset which aims to decompose radiology reports into individual statements (they call this dataset CoVT-CXR). Each statement has an accompanying co-ordinate location, semantic description (such as ‘left 7th rib’), and a textual description of the relevant reasoning required to produce the statement of interest. The author evaluate their framework for VQA and report generation against a number of closed-source models such as GPT-4o and some open-source frameworks such as LLaVa-1.5.

**Strengths:**

- The paper notes an appropriate lack of 'interpretable' medical datasets, and this is true for multi-modal radiological datasets. The introduction of CoVT-CXR would be a welcome resource for a number of researchers in the medical AI field.
- The 5-stage 'step decomposition' framing is intuitive and easy to both motivate and understand.
- The authors attempt a number of ablation experiments to demonstrate both the utility of multi-step reasoning, and the fact that additional traces are associated with improved performance.

**Weaknesses:**

I have a number of questions and comments that relate to the quality of this work.
- Whilst the CoVT-CXR dataset does sound useful, it is not possible for me to review any example instances of it. The authors explain all code and the dataset will be released upon acceptance. However, as the dataset is framed as a core contribution, it would be more useful If reviewers could review both an anonymous codebase of their experiments AND at least some instance of the proposed CoVT-CXR dataset.
- This approach appears to be inference-heavy. Visual and textual information may need to be passed to the model upwards of 12 times to perform inference for a statement $\mathcal{S}$ (as per the abstract - Line 26). Could the authors comment on this? If this is a limitation, it should probably be acknowledged in the main script.
- If the dataset required 32 medical professionals and a bespoke labelling platform, it is unlikely to be a scalable data labelling approach. What it I wanted to do this for brain MRIs, for instance? Unless the authors have a different view, this should also probably be acknowledged as a limitation.
- How did you deal with incomplete or incorrect ground-truth radiology reports in MIMIC-CXR?
- How were the 112 semantic classes defined?
- Segmentation strategy is quite unclear from lines 271-275. Are you using pretrained segmentation models with prompt engineering to ‘enable zero-short capability for medical images’, or are you fine-tuning models with CXRs, or something else? → if using a generic segmentation model, how do you account for poor segmentation results?
- How exactly does the framework autonomously identify the report all relevant targets within the images (line 325)? If inference is performed multiple times, can we not end up with the same statement being produced more than once? An explanation is not clear in the manuscript.
- The authors do not compare their framework to Llava-Med, MAIRA-1/2, CheXAgent, etc. I.e., there are no comparisons with VLMs specialized for medical tasks/CXRs specifically.
- Only lexical metrics are used for eval. These do not appropriately capture meaning as they treat all terms equally (please see https://arxiv.org/pdf/2406.04449 for more detail) — what about RGER, Chexbert, RadFact, etc? These are well known clinical evaluation metrics that are reported in most radiology report generation research.
- Unfortunately, for both ROUGE and meteor metrics (reported lexical metrics), results are somewhat underwhelming. For instance Phi-3V outperforms the proposed CoVT when finetuned. LLaVA-1.5 (not LLaVa-med) matches or outperforms CoVT for both VQA and report generation when fine-tuned.
- Line 490: Authors claim that incorporating more intermediate steps significantly improves the average sampling probability of ground truth tokens, which enhances the likelihood of generating accurate reports. Could it not simply be the case that incorporating more intermediate steps simply exposes the model to more “ground-truth” tokens, and is thus not a function of improved reasoning per se, but rather something more like oversampling these tokens and therefore generating them more frequently? It would be useful to have clinical metrics rather than purely lexical ones to help make this claim.

**Questions:**

Please refer to the weaknesses above.

In addition to the above points, the paper appears to have not been appropriately spelling- or format-checked in many places. Here is a non-exhaustive list:

Paragraph 2:
are over-demanding for massive data → demand massive data.
learning process compared to human → learning process compared to humans.

Fig 1:
There is a mild enlargement of the cardiac silhouette is present → Mild enlargement of the cardiac silhouette is present

Line 92-93:
generic as there are all structures and major types of pulmonary pathologies included → generic as all [anatomical] structures and major pulmonary pathologies are included.

Fig 2:
for one “S” in red? What is “S” explicitly here? Presumably a sentence/segment → if so, is every sentence of every report annotated in this way?

Line 131-132:
shed lights → shed light

Line 150:
report segment → report segments

line 152:
corrdinate space → co-ordinate space

Fig 4 T1 panel:
<classe> → <classes>

Line 380:
as shown in 5. → as shown in Fig. 5.

Lines 471-472:
formatting error — clash between text and Table 3 caption - likely due to vspace adjustment.

---

> ### Comment · Reviewer_yTXu · 2024-11-25
>
> I would like to thank the authors for their work. I am grateful they have added some clinical evaluation metrics, which I believe to be crucial for any paper investigating radiology report generation.
>
> I have two major points based on their rebuttal:
>
> 1. There are quite large score discrepancies from what's quoted in the works themselves. For instance, you quote a Macro-F1-14 performance for MAIRA-2 of **26.63**, whilst in the [original paper](https://arxiv.org/pdf/2406.04449) this metric is quoted as **41.6** (Table D.1). I'm quite surprised by the vast difference in quoted scores for this metric. Did you use the MIMIC-CXR test-split? If you did, could you please comment on/explain this discrepancy?
>
> 2. I still don't understand one of the most important steps of this pipeline. I read main l.271-275 once more, as you recommended, and actually the explanation is exactly the same as before. It says "even models fine-tuned with medical images struggle to capture overlapping structures like bones. This further motivates us to explore [the] SAM framework. In practice, we employ models trained for T1 to support early metadata curation."
>
> Here, "T1" is your first task - which is to segment all possibly relevant structures in a CXR. Given it's the very first thing you do, I would argue the entire quality of the pipeline to some extent is bounded by how well this step works. I'd please like a clear explanation of which models are being used here. You're saying 'even fine-tuned models' for medical imaging did badly, but then you employed models "trained for T1"? Do you mean something like this?: https://arxiv.org/abs/2411.03064 If not, what?
>
> I also have one minor point:
>
> The paper still contains a few typos. For instance, in the sentence I just quoted above regarding the segmentation strategy. Please do try to read the main manuscript thoroughly and correct errors.

---

> ### Author Response · Authors · 2024-11-25
>
> Thank you for the reviewer’s timely feedback.
> - The discrepancy of MAIRA-2 performance:
> > 1.	The MIMIC-CXR dataset includes an overwhelming number of normal cases. During annotation, we realized that diagnostic reasoning chains for these cases offer limited value. Therefore, we did not use the original dataset split and instead focused on more complex cases to better highlight multimodal reasoning chains, which leads the discrepancy of MAIRA-2 performance.
> - SAM-CXR building:
> > 2.	We trained a SAM model for CXR from scratch using a manually annotated dataset. Existing models, including SAM, MedSAM, and the lung-focused SAM model suggested by the reviewer, did not provide adequate visual recognization for CXR. (main l.274-279)
> - Revised manuscript:
> > 3.	We have addressed all the mentioned typos, incorporated additional related work such as Maira-1/2, and included a limitations section (main l.523-533). To adhere to ICLR’s 10-page limit, the visualization results have been moved to the supplementary material.
> - Clarification of the “S” notation (mentioned in typos):
> >4.	In Fig. 2, “S” represents a disentangled sentence from the report. CoVT-CXR performs reasoning on a sentence-by-sentence basis to avoid identifying multiple independent targets within a single sentence (main paper lines 156-160).

---

### Meta-Review · Area_Chair_iEdr · 2024-12-21

**Metareview:**

This work introduces CoVT-CXR, a chain-of-visual-thought framework for interpretable chest X-ray diagnosis, supported by a meticulously annotated dataset that facilitates multi-step reasoning aligned with clinical workflows. Strengths include the innovative methodology that mimics medical reasoning, the introduction of a unique dataset and annotation tool for fine-grained diagnostic reasoning, and strong experimental results demonstrating the approach's effectiveness compared to existing methods. Weaknesses raised include limited scalability of the manual annotation process, dependency on inference-heavy models, and a lack of comparisons to certain specialized medical models; however, these were addressed with detailed clarifications, updated metrics, and plans for future work.

**Additional Comments On Reviewer Discussion:**

The reviewers highlighted concerns about scalability, dataset clarity, comparisons to specialized medical models, and the reliance on general lexical evaluation metrics. The authors addressed these by providing additional clinical evaluation metrics, clarifying the dataset creation process and annotation methodology, and incorporating comparisons to medical-specific models like Med-LLaVa and CheXagent. They also acknowledged limitations in scalability and proposed solutions such as semi-automatic annotation tools for future dataset expansion. One reviewer has raised the score.

---

### Decision · Program_Chairs · 2025-01-22

Reject